# Continuous Lactate Monitoring System Based on Percutaneous Microneedle Array

**DOI:** 10.3390/s22041468

**Published:** 2022-02-14

**Authors:** Ming-Nan Chien, Shih-Hao Fan, Chi-Huang Huang, Chien-Chen Wu, Jung-Tung Huang

**Affiliations:** 1Department of Mechanical Engineering, National Taipei University of Technology, No. 1, Sec. 3, Zhongxiao E. Rd., Da’an Dist., Taipei City 10608, Taiwan; chienmingnan@gmail.com (M.-N.C.); benny5992211168@gmail.com (S.-H.F.); 2Division of Endocrinology and Metabolism, Department of Internal Medicine, MacKay Memorial Hospital, Mackay Medical College, Taipei City 25245, Taiwan; 3Graduate Institute of Sports Science, National Taiwan Sport University, No. 250, Wenhua 1st Rd., Guishan Dist., Taoyuan City 33301, Taiwan; huang@ntsu.edu.tw; 4AREDOT Smart Kinetic Energy Co., Ltd., 3rd Floor, No. 82, Chang’an W. Rd., Datong Dist., Taipei City 10355, Taiwan; aredot.jackie@gmail.com

**Keywords:** lactate, biosensor, three electrode system, microneedle array, continuous monitoring, cyclic voltammetry

## Abstract

Lactate measurement is important in the fields of sports and medicine. Lactate accumulation can seriously affect an athlete’s performance. The most common problem caused by lactate accumulation in athletes is muscle soreness due to excessive exercise. Moreover, from a medical viewpoint, lactate is one of the main prognostic factors of sepsis. Currently, blood sampling is the most common approach to lactate measurement for lactate sensing, and continuous measurement is not available. In this study, a low-cost continuous lactate monitoring system (CLMS) is developed based on a percutaneous microneedle array that uses a three-electrode lactate sensor. The working electrode has an area of 10 mm × 6 mm, including a 3 × 3 array of stainless-steel microneedles. The length, width, and thickness of each needle are 1 mm, 0.44 mm, and 0.03 mm, respectively. The working electrode is then plated with gold, polyaniline, lactate enzyme, Nafion, and Poly(2-hydroxyethyl methacrylate) (poly HEMA). The reference electrode is a 2 × 1 array covered with AgCl, and the counter electrode is a 2 × 1 array plated with gold. The sensor is incorporated into the CLMS and connected to a smartphone application and the cloud. The CLMS was tested on 40 human subjects who rode indoor bicycles, starting at 100 W and increasing in steps of 25 W at intervals of 5 min until exhaustion. The data acquired from the app connected to the CLMS were analyzed to determine the subjects’ lactate response to exercise and the feasibility of assessing exercise performance and training exercise intensity by using the proposed system.

## 1. Introduction

Lactate sensing plays a dominant role in predicting the performance of athletes, and currently, it is mainly performed using blood sampling analyzers [1,2]. Although such measurements are accurate, users are required to stop moving during the measurement for blood collection from their fingertips or earlobes. This measurement method is uncomfortable for athletes, cannot be used for continuous measurement, and is relatively expensive. Lactate measurement is important in medicine as well. Under normal circumstances, the lactate level of the human body is 0.5–2 mmol/L at rest, and before the onset of sepsis, it increases abruptly to more than 1 mmol/L [3]. Therefore, the continuous measurement of lactate levels is important in both medicine and sports science.

Originally, subcutaneous drug delivery [4] and the detection of the concentrations of various chemicals [5] were mostly performed using microneedles. In recent studies, micro-invasive microneedles have often been used to measure chemical concentrations. Somasekhar R. Chinnadayyala et al. (2021) [6] used a microneedle with the height, width, and length of 650 μm, 110 μm, and 150 μm, respectively, to develop a minimally invasive continuous blood glucose monitoring sensor based on a non-enzyme method. In addition, they used cyclic voltammetry and electrochemical impedance methods to conduct subdermal experiments on mice [7,8]. After the microneedles were implanted under the skin of the mice, the device worked accurately for approximately 10 days [9], after which the sensor measured the signal incorrectly. 

In addition, Joshua Ray Windmiller (2011) [10] and others used a 3 × 3 array of hollow microneedles to measure lactate levels. Hollow microneedles are helpful for loading several biocatalysts. Paolo Bollella et al. (2019) [11] used a 4 × 4 array of hollow microneedles to measure lactate levels. In their research, the surfaces of the microneedles were treated specifically to achieve high measurement accuracy. The measurement instruments used in the two abovementioned experiments are commercially available to date, meaning that self-developed sensor circuits need not be used. Moreover, in vitro experiments remain the mainstay, and measurements on human subjects have not yet been carried out. For this reason, the actual performance of the microneedles after entering the human body is unknown.

The normal range of lactate in human blood is 0.5–2 mmol/L. A change in blood lactate concentration is an important and sensitive indicator in medical conditions such as shock, blood disorders, and sepsis, which can be diagnosed and assessed by measuring the lactate level in the blood. Many detection methods are affected by variability and low sensitivity. The enzyme color determination method that uses lactate oxidase (LOX) has been adopted in clinical laboratories, but because enzymes are expensive, it is not feasible to perform multiple experimental verifications. In summary, the LOX biosensor is superior to the lactate dehydrogenase (LDH) enzyme owing to its advantages of simple response and easy production [12].

The contributions of this paper are as follows: We have developed a continuous lactate monitoring system that employs a microneedle array instead of a single microneedle and have attempted to demonstrate its feasibility as a lactate sensor. The microneedle is fabricated by following a low-cost stamping process. Moreover, because the microneedle dimensions are length—1 mm, width—0.44 mm, and thickness—0.03 mm, its insertion into the human body is minimally invasive and almost painless. After the completion of the development of the sensor, we tested it for continuous lactate measurement on 40 human subjects. The exercise selected for testing was indoor bicycle riding. The sensors were attached on both thighs and calves of the subjects. The subjects were instructed to exercise by starting to ride at 100 W, increasing their intensity by 25 W at intervals of 5 min, continuing until exhaustion, and resting thereafter. The continuous lactate sensor developed in this study achieves the goals of being minimally invasive, bloodless, and continuously monitoring (see Figure 1).

## 2. Materials and Methods

The sensor developed herein uses a circuit board with a sensing function and cyclic voltammetry to obtain measurements from a three-electrode electrochemical sensor. The three electrodes are the working electrode, reference electrode, and counter electrode.

First, the circuit board is connected to the percutaneous microneedle of the three-electrode electrochemical sensor, which is attached to a patch and then penetrated under the skin to measure the lactate concentration in the subcutaneous tissue fluid. The measured lactate concentration is transmitted to a smartphone app through the Bluetooth module on the circuit board and displayed so that the user can observe their own lactate value over time. The value displayed on the mobile phone is synchronized to the cloud to facilitate data collection and analysis.

In what follows, we describe the mechanism of the developed sensor, film formation on the percutaneous microneedle, and design of the microneedle array.

### 2.1. Introduction to Sensor Development and Instrumentation

This section introduces the development and rationale of the sensor, as well as the chemistry used therein. The chemical materials used in this study are as follows: polyaniline (RichHealth Technology Corporation, Hsinchu, Taiwan), Nafion (Uni-Onward Corp., New Taipei City, Taiwan), poly(2-hydroxyethyl methacrylate) (poly-HEMA) (Uni-Onward Corp.), phosphate buffer (PB) (Uni-Onward Corp) (pH = 7.2), l-lactic acid (FunYen Technology Inc., Hsinchu, Taiwan), and lactate oxidase (Toyobo, Japan).

#### 2.1.1. Range of Lactate Concentration in Human Body

The main energy sources fueling human exercise are aerobic and anaerobic metabolism. The process of metabolism produces lactate and hydrogen ions. If the lactate and hydrogen ion decomposition rate of the body is slower than their production rates in the metabolic system, lactate and hydrogen ions accumulate, causing muscle soreness and fatigue.

The value of lactate in the blood of a normal person without exercise is approximately 0.5–2 mmol/L [13]. In the experiment conducted herein, the target sample for measurement using the sensor was subcutaneous interstitial fluid, and its lactate value was 1.9–2.2 mmol/L. The actual lactate values under different conditions are summarized in Table 1.

#### 2.1.2. Cyclic Voltammetry

During scanning from a low potential to a high potential, the analyte generates an oxidation peak corresponding to an oxidation current. CV diagrams can help us to determine the potential at which the oxidation reaction occurs [14].

The circuit board used in this study employs cyclic voltammetry to analyze the lactate concentration by determining the oxidation current corresponding to the lactate concentration. In all the CV methods in our experiments, the position of 0.35 V is taken to read the current value.

Figure 2 shows a schematic diagram of the CV method for lactate measurement in this experiment using a PalmSens potentiostat in conjunction with our fabricated microneedles. The parameters of PalmSens are as follows: E begin = −0.4 V, E vertex = 0.6 V, E step = 0.001 V, and scan rate = 0.06 V/s.

#### 2.1.3. Three Electrode System

The three electrode system is commonly used for lactic acid detection, and the measurement method of the sensor fabricated in this study is also the same. The constituent elements include a working electrode (WE), counter electrode (CE), and a separate reference electrode (RE). The working electrode can be considered an electron donor and receiver under an appropriate applied voltage in an electrolyte environment. The working electrode should preferably have the characteristics of a high signal-to-noise ratio and a wide potential window range. It is generally used in electrochemistry. The major working electrode materials used in the field are carbon, mercury, platinum, and gold.

#### 2.1.4. Percutaneous Microneedle

The working electrode used for lactate detection is a microneedle sheet composed of a stainless-steel stamping. The stamping process reduces the production cost considerably. The needle tip is plated with gold. The working electrode has an area of 10 mm × 6 mm, including a 3 × 3 array of microneedles, where the length, width, and thickness of each needle are 1 mm, 0.44 mm, and 0.03 mm, respectively. The working electrode is then plated with gold to facilitate the subsequent adhesion of materials, including polyaniline, lactate enzyme, Nafion, and poly HEMA. The reference electrode is a 2 × 1 array covered with AgCl, and the counter electrode is a 2 × 1 array plated with gold. Figure 3 shows a picture of the percutaneous microneedle array used as the working electrode, and Figure 4 shows a schematic diagram of a microneedle array that has pierced the skin. The porosity of the polyaniline (PANI) layer can be observed on the surface of the microneedle using a scanning electron microscope (SEM, JSM-7610F, Jeol), as shown in Figure 5a. The porosity of PANI is important because the more pores there are, the more enzymes can be adsorbed onto it. To ensure that the conditions are conducive for the microneedle to pierce the skin, the tip of the microneedle is sharpened, as shown in the SEM image in Figure 5b.

#### 2.1.5. Principles of Electrochemistry

L-lactate can be electrochemically measured in terms of the LOX activity. The de-tailed reaction formula is presented in Figure 6. We have applied biosensors based on the electrochemical reaction to determinate the lactate concentration. In the first step of the reaction, lactate reacts with the oxidized form of the enzyme LOX, which converts it into pyruvate accompanied by the reduction of LOX. We used LOX and our electrochemical lactate sensor is based on monitoring the hydrogen peroxide released by the reduction of oxygen as an electron acceptor, which is similar to the function of a first-generation glucose sensor [15].

Lactate oxidase (LOX) is fixed on the electrode surface to detect lactate, and it reacts electrochemically with L-type lactate directly according to the principles of applied potential and oxidation current. Lactase can catalyze the formation of pyruvic acid and hydrogen peroxide through an oxidation reaction in an oxygen-containing L-lactate environment and then, separate oxygen, hydrogen ions, and electric ions through hydrogen peroxide, as illustrated in Figure 6.

#### 2.1.6. Polymer Film-Modified Electrode

Thin film coating is commonly used in working electrode electrochemistry. The commonly used high-molecular-weight polymers include Nafion, Tosflex, and polyvinylpyrrolidone (PVP). Nafion is used as the high-molecular polymer in the working electrode of the sensor fabricated herein because Nafion has excellent ion-exchange properties and biocompatibility, and it can be used to suppress interference.

A polymer film can prevent external interference during measurement and improve the specificity and sensitivity of measurement. Therefore, in this experiment, we required a polymer film for an image current with high biocompatibility and high oxygen permeability. To this end, we have used poly HEMA as the polymer protective film material for the working electrode owing to its high biocompatibility and high oxygen permeability.

#### 2.1.7. Design of Electrode Plating Materials Used for Percutaneous Microneedle Array

To cover the base layer of the percutaneous microneedles, we have used an electroplated conductive polymer film (PANI) to enhance the reaction signal of the enzyme. As the second layer, we have used lactic oxidase (LOX) for redox reactions with lactate. The third layer is composed of a polymer film (Nafion)-modified electrode that can improve adsorption of the object to be detected on the electrode surface. The fourth layer is a polymer film (poly HEMA) for protecting the electrode; this layer isolates impurities that can cause interference. The schematic diagram is shown in Figure 7.

### 2.2. Assembled Device and Circuit Fabrication

In this section, we introduce the fabricated sensor housing and circuit, in addition to describing the final circuit integration. We employed three-dimensional (3D) printing to fabricate the housing and an Altium Designer to design the circuit.

#### 2.2.1. Design of Continuous Lactate Sensor Mechanism

The sensor mechanism consists of four parts: the microneedle array, housing, circuit, and battery box. The microneedle array is composed of a working electrode microneedle array, a reference electrode microneedle array, and a counter electrode microneedle array. The three-electrode arrangement of the microneedle array is shown in Figure 8.

The housing system contains the three-electrode microneedle arrays, an upper cover, a circuit board housing, a lower cover, and a battery holder. The upper cover, circuit board shell, and lower cover are fabricated by 3D printing (ATOM A202083K). The battery holder is commercially available, and it houses two 3.3 V batteries connected in parallel. An exploded view drawing of this assembly, created using SOLIDWORKS, is presented in Figure 9a, and a photograph of the actual assembly is shown in Figure 9b.

#### 2.2.2. Design of Continuous Lactate Detection Circuit for Percutaneous Microneedle Array

The system uses the ELSRA CC2640R2F low-power Bluetooth module [16], which is responsible for controlling the circuit, reading the signal, and transmitting the read data to the device through Bluetooth communication. In the circuit layout, the Bluetooth antenna should be close to the periphery of the circuit board, and no copper should be laid on the upper and lower layers of the antenna area to improve its transmission quality. Electrochemical measurement is extremely demanding, and noise causes interference with the processing and signals. For this reason, in addition to placing bypass capacitors on each power supply terminal to filter out noise from the power supply, we focused on preventing interference between analog and digital signals during the layout. The main signal is provided through holes on both sides to reduce the capacitive effect between the upper and lower layers of the board.

The circuit applies voltage to induce a redox reaction at the working electrode. This charge enhances the current response of the electrode, and finally, the sensing circuit amplifies the electrode’s signal. The counter electrode (CE) is at a negative voltage relative to the working electrode. In addition, the current generated at the working electrode pin is less than 100 nA for mg/dL concentration. Because this current is inadequate, a transimpedance amplifier (TIA) with an extremely low bias current is required to convert the current into an output voltage. To this end, we adopted the MAX9913 operational amplifier, which is suitable for similar applications. It has a low bias current at room temperature and good anti-noise interference characteristics. Figure 10 shows the front and back of the circuit of this amplifier, and Figure 11 shows its circuit design.

### 2.3. Smartphone Application for Percutaneous Microneedle Continuous Lactate Detection

In this study, we use a circuit to apply a potential of 0.35 V, read the current of the app, convert the current signal into a lactate concentration signal, and transmit it to the mobile app.

The mobile phone app used in this research was developed on the Android platform, and Android Studio was used to write the underlying program [17]. The actual user interface is shown in Figure 12. The upper columns from left to right display the temperature, time, and signal status (whether a normal signal is received). The black area in the center is the lactate display interface, which outputs a continuous line graph; the user’s name, height, and weight can be inputted in the respective fields below it.

The options at the bottom allow users to control the app. The method of using these options are shown in the flowchart in Figure 13. After opening the app, click the SCAN option to start the Bluetooth broadcast to connect to the lactate sensor. Then, click the lactate option (ADC in the flowchart). Thereafter, click the temperature option (TMP in the flowchart).

Finally, click the ONCE option (NIF in the flowchart) to visualize the line graph of the continuous lactate needle test.

## 3. Experiment

To study the feasibility of the proposed continuous lactate sensor, we conducted five experiments in this study.

First, we conducted an in vitro three-electrode test to ensure that the electrode manufactured in this experiment can detect changes in lactate concentration. Second, we conducted a sensor performance experiment to observe differences in the measurements of the sensor microneedles before and after skin penetration. Third, we performed a sweat influence test to determine whether sweat affects the measured data and formulate a solution to prevent data interference due to sweat. Fourth, we determined the influence of temperature on lactate measurement by this sensor.

Finally, we conducted a human experiment in collaboration with the National Sports University for research on athletes’ lactate levels by using cycling as the exercise of choice.

### 3.1. Three-Electrode Test In Vitro

To understand whether the manufactured working electrode can accurately detect lactate levels, in this experiment, we used commercially available lactate mixed with PB (pH = 7.2) to realize five different solutions with lactate concentrations of 0 mg/dL, 5 mg/dL, 30 mg/dL, 100 mg/dL, and 200 mg/dL. The commercially available three-electrode detection system PalmSens was used, and the three electrodes were clamped and placed in electrolytes with different lactate concentrations for measurement. The upward and downward concentration measurements ranged from 0 mg/dL to 200 mg/dL and 200 mg/dL to 0 mg/dl, respectively. Two working electrodes were used to conduct these measurements. The parameters of PalmSens were as follows: E begin = −0.4 V, E vertex = 0.6 V, E step = 0.001 V, and scan rate = 0.06 V/s.

After about twenty experiments and microneedle fabrications, we learned that the measurement range of the fabricated sensor is 5–360 mg/dL. Moreover, the measurement variation of each fabricated microneedle is 4–10 μA. When the concentration is 0, because PBS has impedance, when we apply voltage to PBS, we still get a small current value. The results of the experiment are shown in Figure 14. The working electrode in this study effectively measured and distinguished different lactate concentrations. The service life of the sensor is approximately one month.

### 3.2. Sensor Effectiveness Experiment

In order to confirm that the sensor we fabricated was penetrated subcutaneously enough to measure the lactate in the interstitial fluid, we designed the following experiment. As shown in Figure 2, when 0.35 V is applied, the maximum current response will be obtained, so the circuit we made is also applied with 0.35 V and measured by the chronoamperometry (CA) based on the amperometric detection method. In this experiment, after placing the microneedle under the skin, the circuit was applied with 0.35 V for four seconds and 16 data were collected to observe the current value. When no interstitial fluid was contacted or no lactate was measured, the measured current was abnormal and continued to increase, as shown in Figure 15, due to invalid subcutaneous penetration. As can be seen in Figure 16, when the subcutaneous penetration was valid, the microneedles were immersed in the interstitial fluid, and the lactic acid level was measured, the response of current was normal and continued to decrease. The difference between “offline” and “online” is displayed in the app.

### 3.3. Human Body Sweat Influence Test

To understand whether sweat affects the function of the lactate sensor, the lactate sensor was first directly penetrated under the skin, and a measurement was started. After approximately 40 min, it was found that the measured data started to be affected by sweat, and noise was generated, as shown in Figure 17. When a user sweats excessively, sweat may flow from the skin into the microneedle, leading to the generation of an abnormal sensor signal. To solve this problem, we asked a user to spray an antiperspirant on his skin. Thereafter, we re-penetrated the lactate sensor under the skin and commenced the measurement. After approximately 40 min, we found that the measurement data were not affected by sweat, as illustrated in Figure 18.

The antiperspirant used herein was sprayed on the skin, not on the microneedles. And the experiment was started 30 min after spraying the antiperspirant. When spraying the antiperspirant, we tried to avoid the spot at which the microneedle was placed.

### 3.4. Temperature Influence Experiment

Temperature affects the data measured using the fabricated sensor. According to Ziru Jia et al. (2019) [18], when the sensor temperature increases by 1 °C, the measured current value $I_{1}$ is approximately the original current value $I_{0}$ multiplied by 1 plus or minus sensitivity $a_{0}$ multiplied by the temperature difference ΔT, as given in Equation (1):(1)$$I_{1}=I_{0}\times \left(1\pm a_{0}\times \Delta T\right),$$

A temperature calibration experiment was conducted for the sensor developed herein. In this experiment, lactate solutions of the same concentration were heated from 32 °C to 36 °C, and the CV method was used to measure the currents at these temperatures. The experiment was performed in triplicate. The experimentally measured $a_{0}$ was approximately equal to 0.1. The calculated value of 0.1 is the average of the results obtained from multiple microneedle experiments. Finally, $a_{0}$ will be incorporated into the firmware for calculation to ensure that temperature will not affect the measurement results.

### 3.5. Human Experiment

This human trial was approved by the Human Research Ethics Committee of Fu Jen Catholic University, project number: C108210. The exercise equipment used in this experiment was an indoor bicycle. The experiment was conducted as follows: Firstly, the lactate sensor was attached to the upper and lower parts of the subjects’ legs. After sensor attachment, the subjects were instructed to commence riding, starting with an intensity of 100 W, gradually increasing in steps of 25 W at intervals of 5 min until exhaustion, and then stopping to rest. Figure 19 shows a photo of a subject riding an indoor bicycle, and Figure 20 shows the actual sensor installation positions.

In this experiment, the lactic acid in the interstitial fluid was measured before the subjects commenced the bike riding exercise. Then, blood lactate was measured to calibrate the interstitial fluid lactate measured using the proposed sensor. After calibration and during the riding experiment, it was found that the lactic acid concentration changed more quickly in the interstitial fluid than in the blood.

A total of 40 healthy adults (19 women and 21 men) were recruited for the human experiment. Table 2 shows the basic data of these subjects.

In the experiment, the blood samples obtained from the subjects were used to analyze the blood lactate concentrations by using a lactate/blood glucose analyzer (Biosen C-Line EKF Diagnostic, Cardiff, Wales, UK). The inter-assay detection variation was ≤1.5%, and the lactate detection range was 0.5–40 mmol/L. Finally, we compared the blood lactate levels with the data obtained using a single-use lactate sensor to verify whether the lactate sensor measurements were accurate.

After the test, it was found that the blood lactate levels correlated positively to the lactate sensor data, as shown in Figure 21 and Figure 22. However, the values yielded by several lactate sensors were lower than the actual blood lactate levels. As shown in Figure 23, it was surmised that the lactate signal on the microneedle sensor would slowly deteriorate when left for an extended period.

## 4. Conclusions

Lactate sensing is an important measurement in the scientific training of athletes to improve their performance. The sensor developed herein was able to successfully measure the lactate value in the subcutaneous tissue of athletes/humans by using a three-electrode method. This study aimed to develop a wearable biomedical sensing device that could be attached to human skin. By using a micro-needle sensor sheet combined with a sensor circuit module and a carrier that can be attached to the arm and leg, lactate level measurement was rendered micro-invasive and bloodless. Data from the continuous lactate sensor can be used to evaluative the lactate response to exercise over a period of time.

The experimental data from humans/athletes obtained in this study using the proposed continuous lactate sensor were compared to the blood lactate levels measured using commercially available sensors, and the two sets of data exhibited positive correlation. However, the sensitivity of the proposed continuous lactate sensor decreased after four months of continuous subdermal measurement. Further adjustments to improve the continuous lactate sensor could lead to its use in practical scenarios involving the continuous observation of lactate levels. The continuous availability of data on lactate levels during exercise could help to adjust the training intensity and improve the performance of athletes.

## Figures and Tables

**Figure 1 sensors-22-01468-f001:**
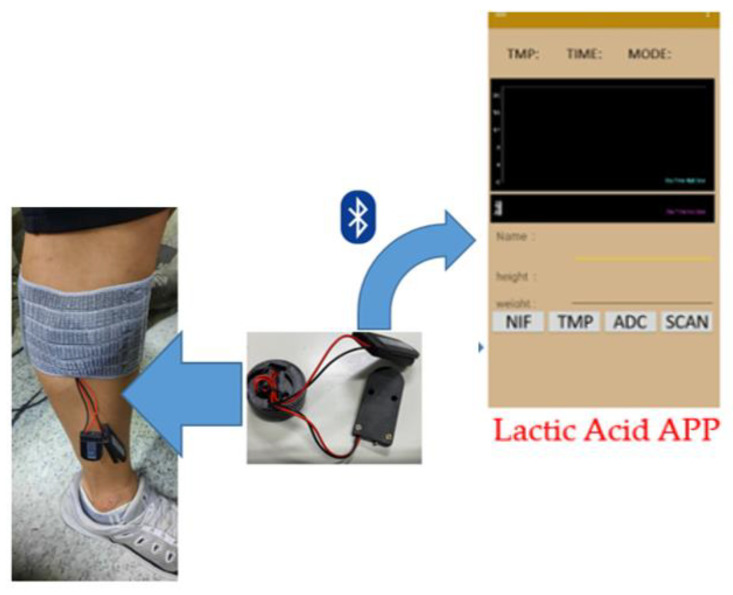
After installation of the continuous lactate sensor under the skin, the Bluetooth function is used to display the lactate data in a specially developed app.

**Figure 2 sensors-22-01468-f002:**
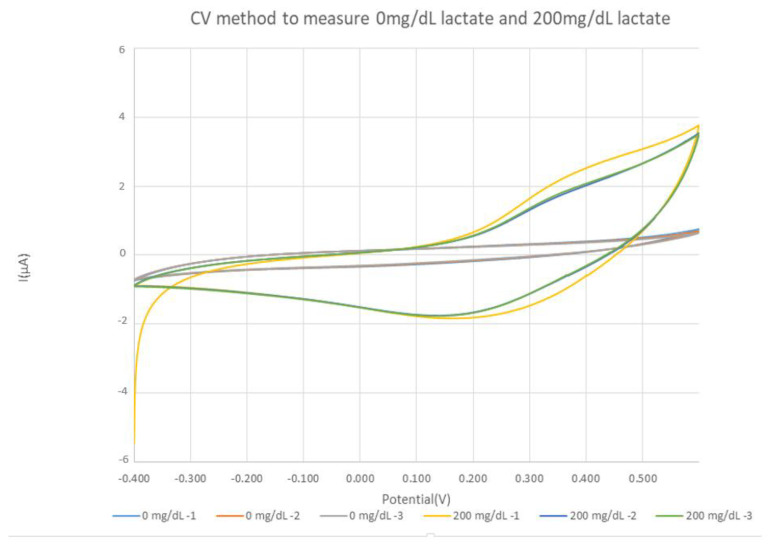
The lactic acid CV map (measured using a PalmSens potentiostat) with a lactic acid concentration of 0 mg/dL and 200 mg/dL.

**Figure 3 sensors-22-01468-f003:**
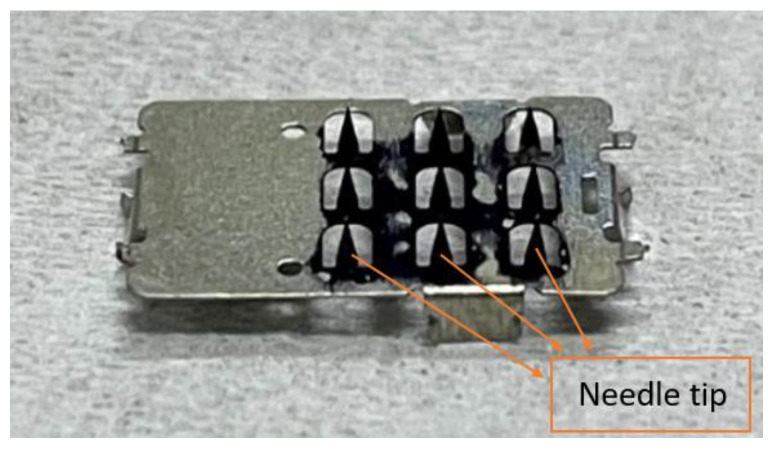
Physical image of the percutaneous microneedle array as a working electrode.

**Figure 4 sensors-22-01468-f004:**
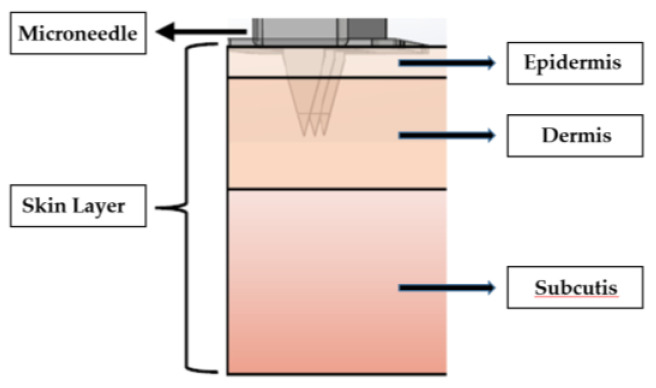
The microneedles penetrate the skin; the length of the microneedles is 1 mm, which is adequate to penetrate the skin and reach the dermal tissue.

**Figure 5 sensors-22-01468-f005:**
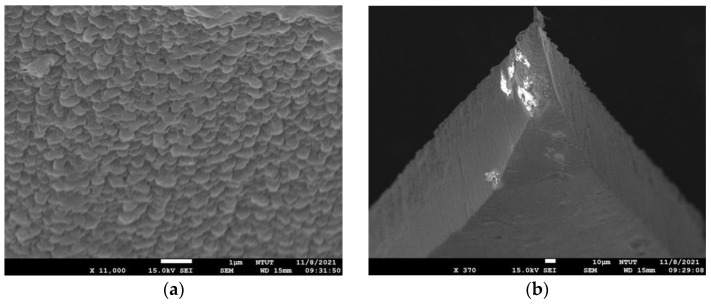
(**a**) The porosity of the PANI polymer layer on the microneedle surface, (**b**) the microneedle sharpening by stamping to ensure that it pierces through the skin.

**Figure 6 sensors-22-01468-f006:**
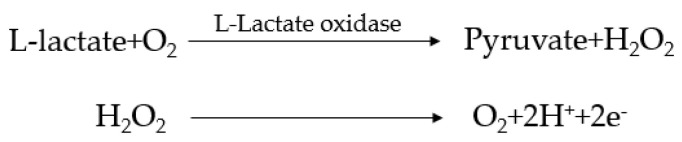
The signal generation scheme.

**Figure 7 sensors-22-01468-f007:**
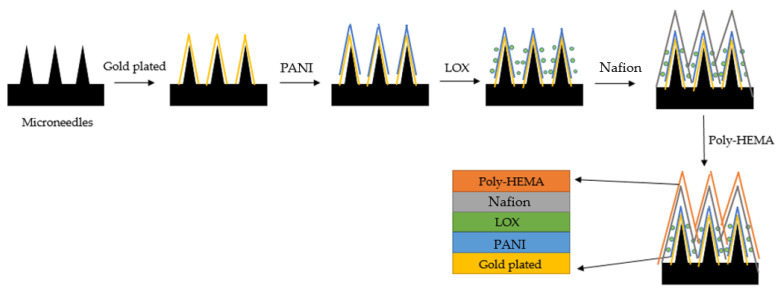
The schematic diagram of materials used to fabricate the percutaneous microneedle array.

**Figure 8 sensors-22-01468-f008:**
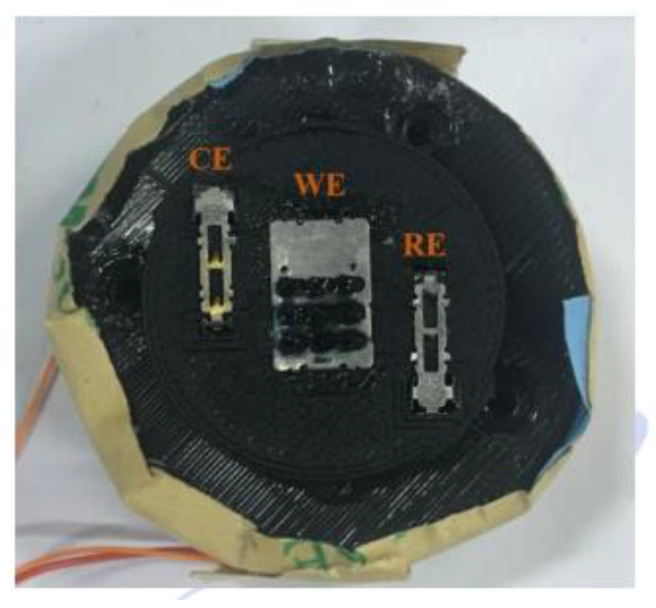
Three electrodes of the microneedle array, including working electrode (WE), counter electrode (CE), and separate reference electrode (RE). The WE is a 3 × 3 metal microneedle array, and the CE and RE are 2 × 1 metal microneedle arrays.

**Figure 9 sensors-22-01468-f009:**
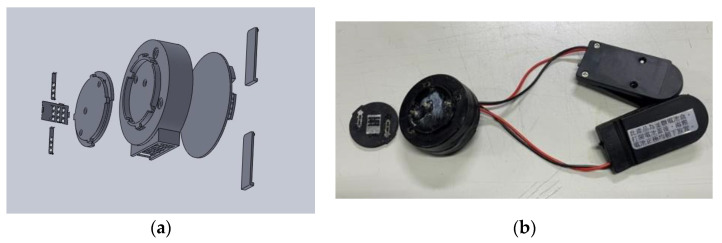
(**a**). The exploded view drawing of the housing system created using SOLIDWORKS: from left to right are the three-electrode microneedle arrays, upper cover, circuit board shell, lower cover, and battery holder. (**b**). The appearance of the assembled sensor.

**Figure 10 sensors-22-01468-f010:**
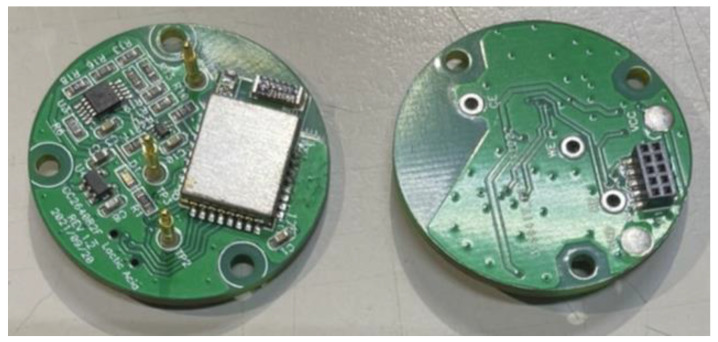
Front and back of the circuit of the MAX9913 operational amplifier.

**Figure 11 sensors-22-01468-f011:**
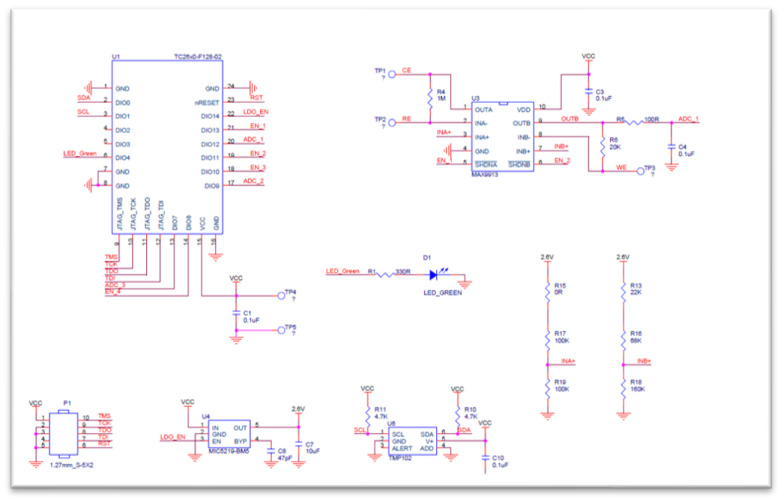
The circuit diagram of the MAX9913 operational amplifier.

**Figure 12 sensors-22-01468-f012:**
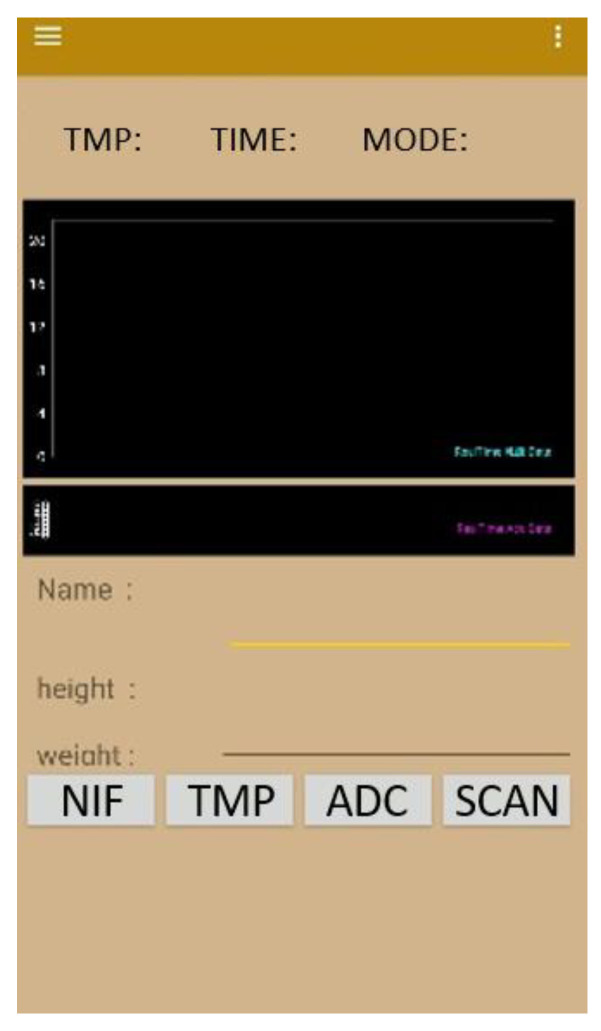
The interface of the Android mobile app for continuous lactate measurement.

**Figure 13 sensors-22-01468-f013:**
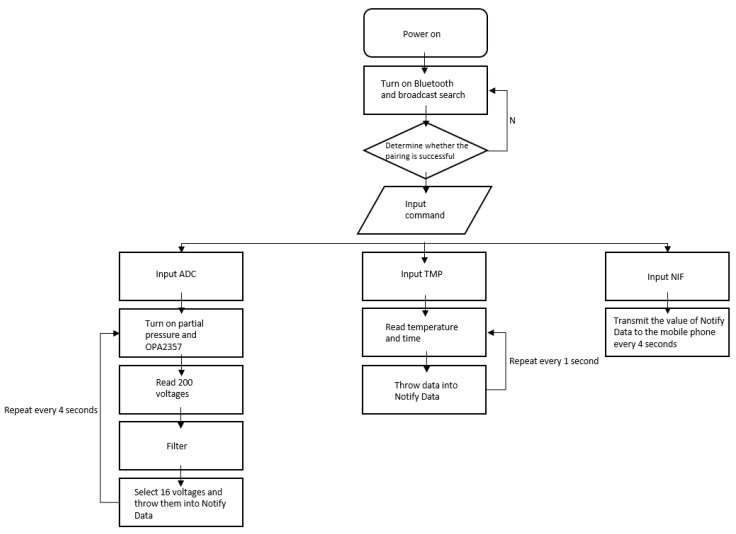
A flowchart showing the Android mobile app.

**Figure 14 sensors-22-01468-f014:**
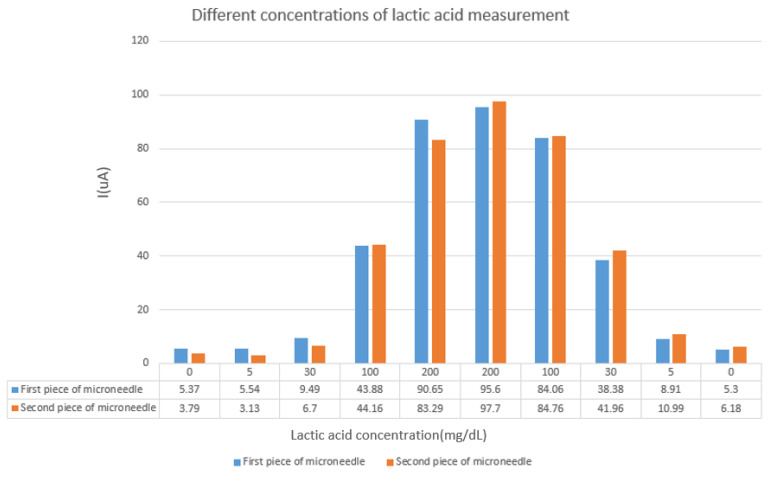
The measurement results with different lactate concentrations on the x-axis and the corresponding currents (µA) on the y-axis.

**Figure 15 sensors-22-01468-f015:**
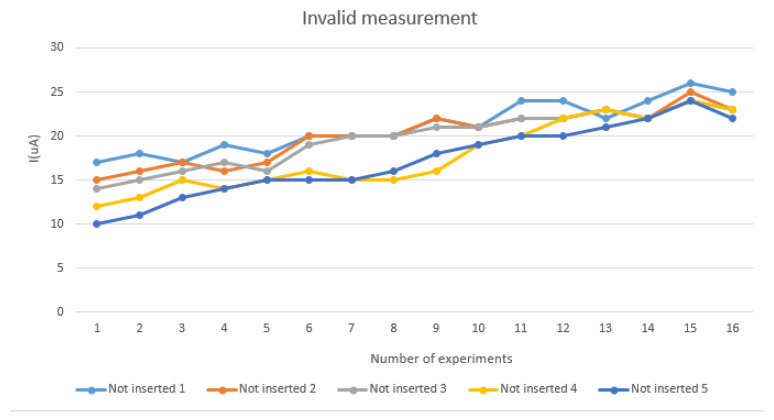
Invalid measurement.

**Figure 16 sensors-22-01468-f016:**
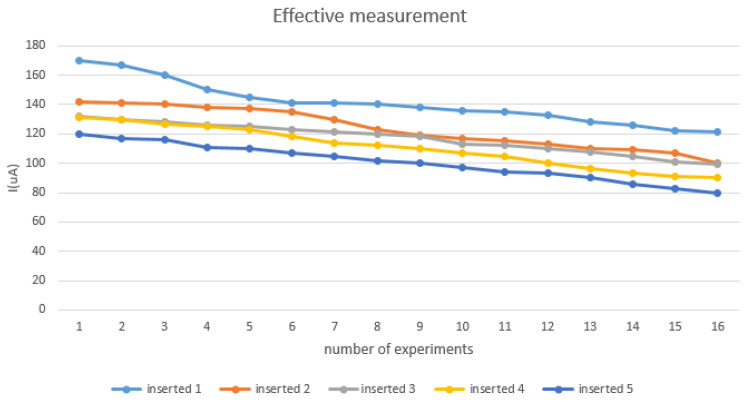
Effective measurement.

**Figure 17 sensors-22-01468-f017:**
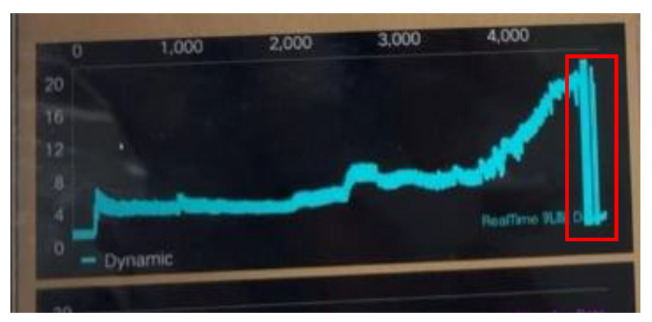
Sweat-affected data.

**Figure 18 sensors-22-01468-f018:**
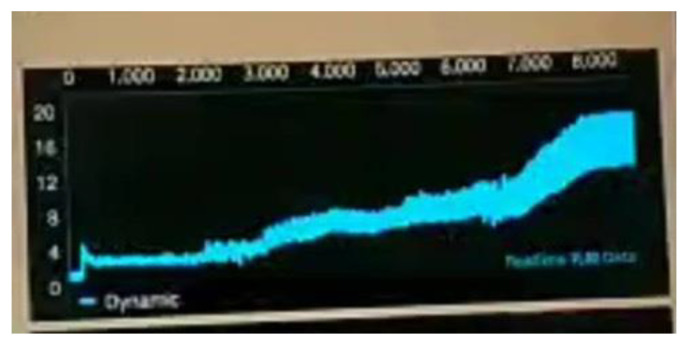
Data unaffected by sweat.

**Figure 19 sensors-22-01468-f019:**
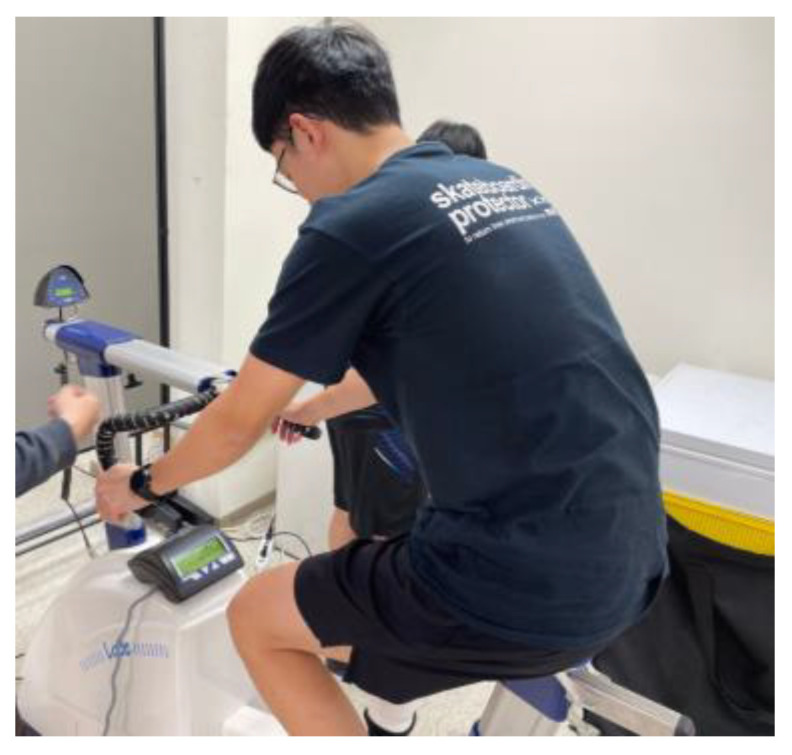
A subject riding an indoor bicycle.

**Figure 20 sensors-22-01468-f020:**
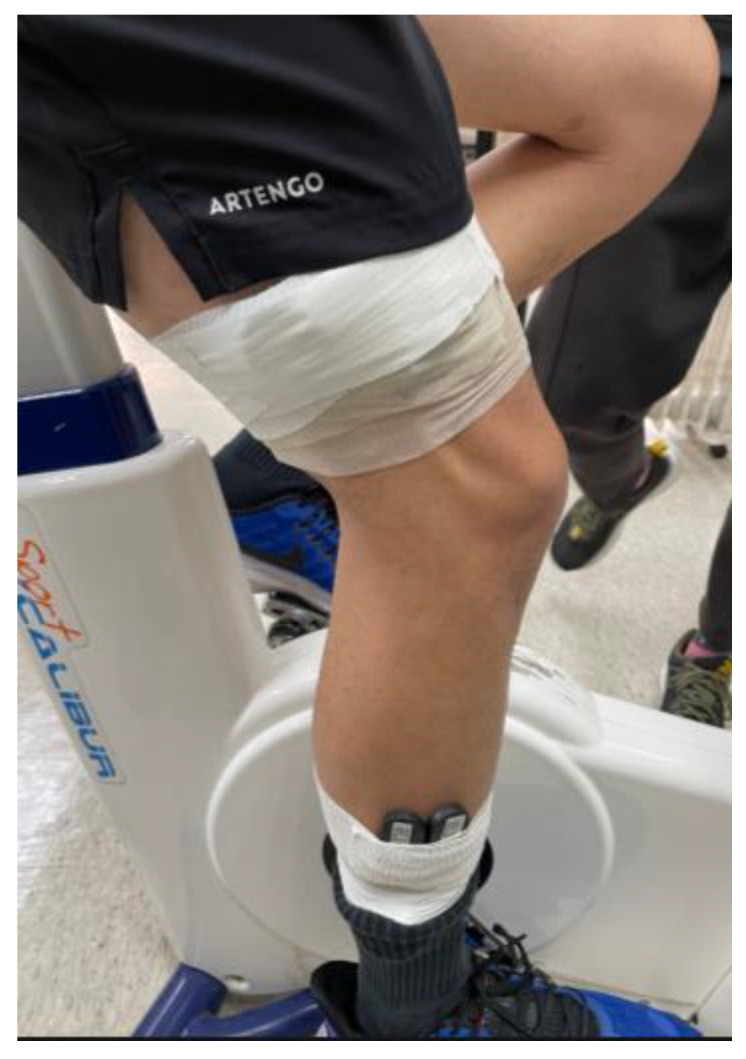
Sensors installed on the upper and lower parts of the leg.

**Figure 21 sensors-22-01468-f021:**
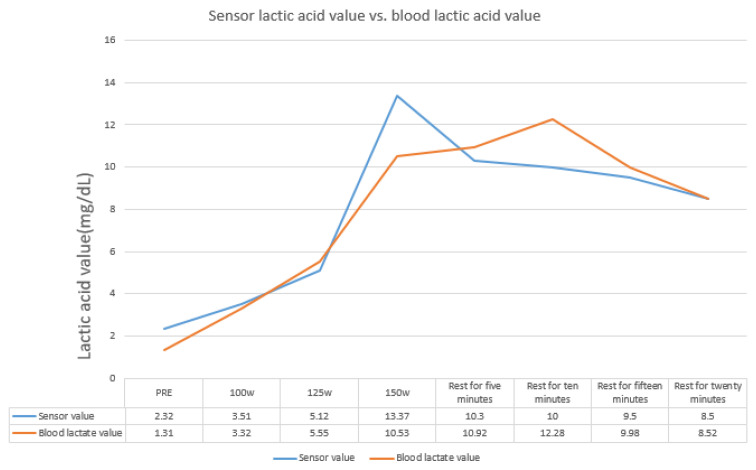
Human experiment data: the data obtained using a lactate sensor show a positive correlation with blood lactate levels.

**Figure 22 sensors-22-01468-f022:**
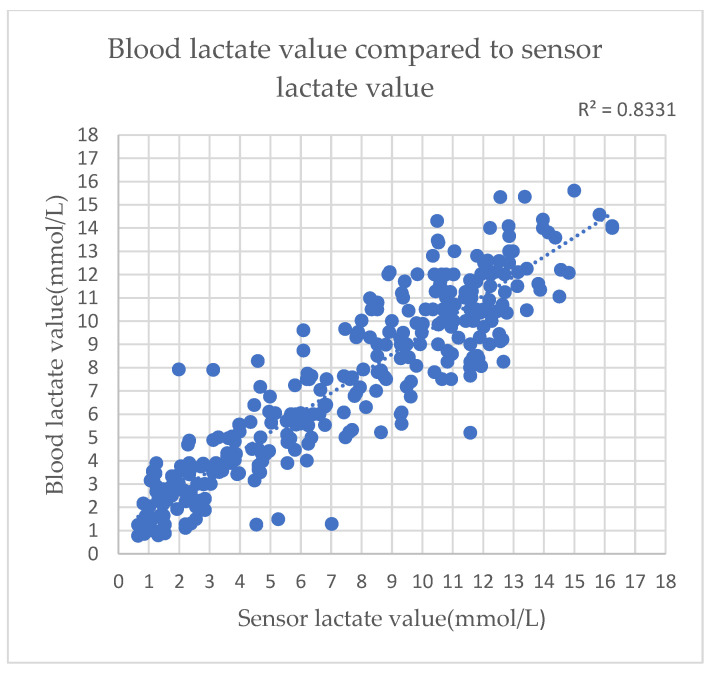
Human experiment data compared to blood lactate levels.

**Figure 23 sensors-22-01468-f023:**
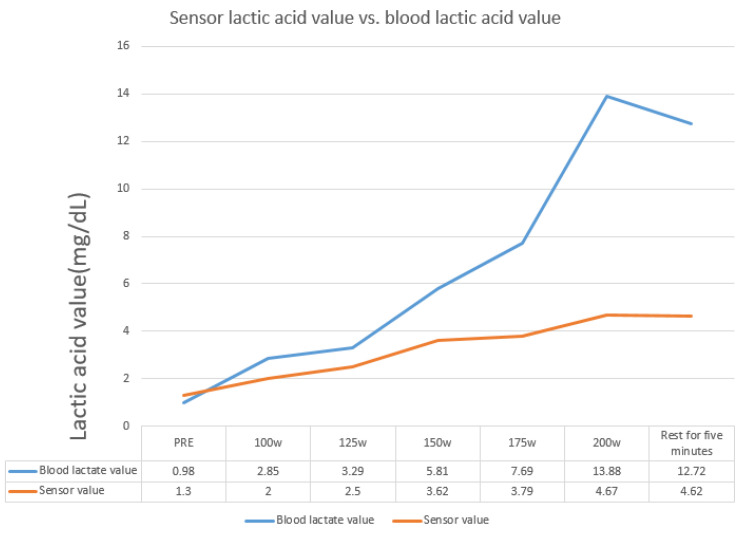
Data from the human experiment measured after placing the microneedle for four months; the results indicate that the enzyme could detect lactate, albeit with a reduced sensitivity to that previously noted.

**Table 1 sensors-22-01468-t001:** The individual status of lactate.

Lactate Status	Normal Lactate	No Exercise	Hyperlactatemia	Excessive Exercise
Detection value mg/dL (mmol/L)	9–39.6 (0.5–2.2)	18–36 (1–2)	>72 (4)	>180 (10)

**Table 2 sensors-22-01468-t002:** The subjects’ basic information, body composition, and thicknesses of thigh and calf sebum.

Project (Unit)	Male (N = 21)	Female (N = 19)
Age	25.38 ± 3.79	25.79 ± 5.27
Height (cm)	173.38 ± 6.04	160.13 ± 6.36
Weight (kg)	68.89 ± 8.4	55.49 ± 6.56
Body fat percentage (%)	16.01 ± 4.92	26.82 ± 5.88
Thigh sebum thickness (mm)	9.1 ± 2.87	14.46 ± 2.59
Calf sebum thickness (mm)	8.05 ± 2.7	10.18 ± 3.24

## Data Availability

The data presented in this study are available from the corresponding author.

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
