# Peer review of "Continuous Lactate Monitoring System Based on Percutaneous Microneedle Array"

_sensors, 2022, doi:10.3390/s22041468_

Round 1

Reviewer 1 Report

Please find attached

Author Response

This human trial has been approved by the Human Research Ethics Committee of Fu Jen Catholic University, project number: C108210.

Below are the replies to the questions:

  • Line 29: delete “was” in “human subjects was who rode indoor” ANS:Revised. 
  • Suggested keywords: lactate, biosensor, microneedle array, continuous monitoring, cyclic voltammetry .  ANS:Revised. 
  • Line 38: correct “and present currently, measurement is mostly done by blood sampling analyzers” ANS:Revised. 
  • Line 41-42: complete sentence “is time-consuming and of relatively high ….. ???” ANS:Revised. 
  • Line 69: LOX and line 71: LDH – acronyms must be defined at first use! ANS:Revised. 

Chapter 2: Material and Methods section. ANS: Revised.

- List of materials is completely missing. Please specify all used materials

- Instrumentation is added here and there. Please add and shortly describe working parameters of all used instruments

- The whole chapter should not have more than 3 sections: 1 - describing the use and working principle of the sensor; and the development of the working electrode (current chapters 2.1, 2.2, 2.3, 2.4, 2.6, 2.7, 2.8, 2.9); 2 – describing the electronics (current lines 89 to 97, chapter 2.5, 2.10); 3 – describing the software (current lines 97-98, chapter 2.11) .

  • Chapter 2.2 -> CVs means scanning from a low potential to a high potential and its reverse scan from the high potential back to the low one. So, there can be both oxidation and reduction processes. If the authors wish to emphasize only the oxidation reaction, they should have used DPVs (differential pulse voltammetry) – it is more sensitive compared to CV. Nonetheless, the authors must add the CV profile of the microneedles in the absence and presence of lactate. ANS: Revised on chapter 2.1.2
  • Chapter 2.3 – a short description of a three-electrode electrochemical system is adequate; however, the authors should review and make some corrections in this chapter: decide and use one naming- counter or auxiliary; simplify the description of the CE - its only role is to pass all the current needed to balance the current observed.at the working electrode; the RE’s role is to act as a reference in measuring and controlling the working electrode potential, without passing any current ANS: Revised on 2.1.3
  • Chapter 2.4 -> The authors make it very difficult to follow on how actually the microneedle array (WE) is actually developed / modified. There is a nice review (Biosensors 2021, 11(9), 296; https://doi.org/10.3390/bios11090296) about microneedle arrays and Fig 1. A has a great schematic illustration on the fabrication process. The authors should add all modification steps in one chapter, and they should adapt figure 8 to the ones in this review in order to make the modification steps clear and easy to follow. ANS:Revised. 

- Definitions for Poly HEMA and PAI are missing ANS: Revised.

- Instrumentation for SEM is missing ANS: Revised.

  • Line 165: information on 3D-printer missing ANS: Revised.
  • Line 177: “with highly specific enzymes” – the authors should mean substrates. The enzyme is LOX which specifically recognizes L-lactate and O2 ANS: Revised.
  • Line 183: what is this oxidized chemical mediator the authors refer to?
  • Figure 7 is correct, however explanations in chapter 2.6 must be revised and corrected! Also, Figure 7 is plagiarised from ref 12, fig 1. Did the authors get permission to use the figure? ANS: Revised.
  • Line 198: define PVP ANS: Revised.
  • Line 227: “The main signal is provided with via holes on” ANS: Revised.
  • Line 229: “This circuit produces an oxidation reaction at the working electrode (WE) during the measurement of lactate” -> a circuit cannot produce an oxidation reaction. This happens either due to an applied potential or due to the chemical reaction at the WE surface due to the presence of lactate. The authors should clarify for themselves how exactly the electrochemical process occurs from the biochemical perspective; they seem to understand the technical part, but not the biochemical implications. ANS: Revised.
  • Chapter 2.11 should also describe how the application reads the output. Initially the authors say that CV is used. Does the software read the maximum of the oxidation peak? Is there only one specific oxidation peak? It is required to add a CV and to explain how the data is read and converted from the CVs. Every 4 seconds there is an output for ADC – usually a CV takes longer than 4 seconds. What is the scan rate used for the CVs? Between which potentials? ANS: Revised.
  • Line 278: define PBS, which pH value? ANS: Revised.pH=7.2
  • Line 280: “The commercially available three-electrode detection system Palmsens was used and the three electrodes ….” – do the authors mean that they connected the developed sensor to a commercially available Palmsens potentiostat? If yes, please provide Instrument information, conditions in which the measurements were performed, CV, between which potentials, scan rate ANS: Revised.
  • Figure 13: define the value on the Y axis. If the measurements are CVs, it should be current … in mA, μA? Also, in the table below, the top row is the actual added value, and the other two rows show the concentration detected by the microneedle. The difference for some of them is quite high. How do the authors explain a measured value of 3-5 mg/dL when the actual concentration is 0? For a few years now, the convention for litre is capital L – correct throughout the whole manuscript. ANS: Revised.
  • Chapter 3.2. -> It is unclear how the measurements were performed, please rephrase this chapter. How can the lactate level increase if the sensor is in air? The enzyme can oxidize in the presence of oxygen, but an electrochemical measurement requires an aqueous media, otherwise the electrodes can be compromised, so please explain what exactly happens. ANS: Revised.

  • Figure14 and 15- define the value on the Y axis. What represents “Number of transactions” on the X axis? ANS: Revised.
  • Line 329: “The experimentally measured a was approximately equal to 0.1.” – this sentence is confusing, 0.1 is the value of the correction, the standard deviation? How was it used. If the authors mention it, it should be better explained.
  • Lines 352-356 – Do the authors state that the developed sensor is actually for single use? Please clarify, comment on this. ANS: Revised.
  • Figure 20. Define units on Y and X axis.
  • Caption Figure 20: the positive correlation is given by the slope from Fig 21. Please delete it from here. What is the value of the correlation coefficient? ANS: Revised, Correlation coefficient=0.8331
  • Figure 22: Define units on Y and X axis. ANS: Revised.

Reviewer 2 Report

The manuscript describes the performance of  continuous lactate monitoring system based on percutaneous microneedle array. This system allows persistent monitoring of lactate controlled by application installed to mobile phone. Although the wearable system described in this manuscript is interesting and promising, chemical part is far from ideal. I cannot recommend this manuscript for publishing in the Sensors journal in the current state. Authors should make some important changes before resubmission:

1) English language should be thoroughly checked through the manuscript text.

2) In reference [3] authors reported of plasma lactate levels > 1 mmol/L for patients with sepsis, in manuscript we can find phrase, corresponding to this reference: “sudden increase of blood lactate to more than 2 mmol/l before it’s onset”.

3) Line 43 and Line 65 duplicate each other. One of them should be removed. Normal lactate level differs from time to time: 1-2 mmol/L (Line 43), 0.5-2 mmol/L (Line 106). Please unify it through the text.

4) Lines 69 and 71: authors introduced LOX and LDH abbreviations without interpretation (the same problem with PAI, polyHEMA). For LOX it appears only in Part 2.6.

5) Authors use cyclic voltammetry to perform measurements with sensor. But there are no one voltammograms in the manuscript. The only demonstrated experimental curve seems to be amperometric. So I’d recommend to clear up this misunderstanding. Also it is strictly necessary to adduce experimental conditions such as potential range, scan rate etc. for cyclic voltammetry or potential for amperometry. Part 2.2. looks very poor.

6) There is a technical mistake in Table 1: range of lactate mentioned as 9-3.96. Authors should correct it.

7)” When the working electrode undergoes an oxidation reaction, the auxiliary electrode undergoes a reduction reaction, and the auxiliary electrode does not participate in the electrochemical reaction but only provides a charge balance function” – meaning is not correct, please change.

8) Area of counter electrode should be at least ten times greater then the area of working electrode. Herein, area of working electrode obviously exceed the area of counter electrode. Authors should explain this point more thoroughly.

9) On Figure 5, counter electrode and reference electrode look like plate with two dark areas, though authors describe it in the manuscript as 3*1 array. Authors should explain this point more thoroughly.

10) From Figure 3 it is not clear: if microneedle penetrates through the dermis till the subcutis or it stays in the dermis? It has been written overall the manuscript: “sensor inserted under the skin”, that seems to be a result of surgery. But at Figures 18 and 19 there are two healthy volunteers with microneedles system applied over the body. Authors should operate with terms more accurate.

11) The presented idea of lactate biosensor microneedle system is not new. There are many similar publications (https://doi.org/10.1016/j.microc.2020.105830, 10.1016/j.snb.2019.127645, 10.1021/acs.analchem.9b05109), that’s why 17 references is not sufficient.

12) Line 199: “Nafion is applied on the enzyme layer to obtain higher electrode surface adsorption and improve electrode detection sensitivity.” Nafion is a polyanion and usually employs in lactate biosensors to exclude the influence of interfering compounds. The disposition of Nafion layer over the recognition layer is also unclear: Nafion layer should repulse the negatively charged lactate anion. Add the explanation of this layer organization.

13) The manuscript describes the performance of continuous lactate monitoring system but there is no mention about its analytical or operational characteristics. There is no determination of concentration range, LOD, lifetime, stability and so on. Lack of this information is inadmissible.

14) What does mean “first piece of microneedle” and “second piece of microneedle” on Figure 13? “Two working electrodes were used for these measurements” can be considered that authors made only two attempts to detect lactate. It is not enough for adequate results interpretation.

15) How did the composition of antiperspirant affect the results? What about contamination problem?

16) If 40 volunteers were tested, why only two results presented in the article? Where the standard deviation for the group?

17) Why do authors not discuss the difference between lactate concentration and its changing rate in blood and subcutaneous tissue fluid? Why is there no information about normal and abnormal levels of lactate in subcutaneous tissue fluid?

18) There is no information about the pH of the working buffer. It might be critical for enzymatic biosensors. PBS buffer abbreviation means Phosphate Buffered Saline solution – please check if NaCl was present in the media in this work.

19) There are some misprints and poor phrases in the manuscript: Line 41 “of relatively high”, Line 29 “CLMS was tested on 40 human subjects was who rode” – second “was” is unnecessary.

20) Figure captions should be extended and contain all the conditions of the experiments.

Author Response

1)English language should be thoroughly checked through the manuscript text.

ANS: We are in consultation with the English editor.

2) In reference [3] authors reported of plasma lactate levels > 1 mmol/L for patients with sepsis, in manuscript we can find phrase, corresponding to this reference: “sudden increase of blood lactate to more than 2 mmol/l before it’s onset”.  

ANS: Revised on line 46.” sepsis results in a sudden increase of blood lactate to more than 1 mmol/l before it’s onset”

3) Line 43 and Line 65 duplicate each other. One of them should be removed. Normal lactate level differs from time to time: 1-2 mmol/L (Line 43), 0.5-2 mmol/L (Line 106). Please unify it through the text. 

ANS: Revised on line 45.” the lactate level of the human body is about 0.5-2 mmol/l”

4) Lines 69 and 71: authors introduced LOX and LDH abbreviations without interpretation (the same problem with PAI, polyHEMA). For LOX it appears only in Part 2.6.

ANS: Revised.

5) Authors use cyclic voltammetry to perform measurements with sensor. But there are no one voltammograms in the manuscript. The only demonstrated experimental curve seems to be amperometric. So I’d recommend to clear up this misunderstanding. Also it is strictly necessary to adduce experimental conditions such as potential range, scan rate etc. for cyclic voltammetry or potential for amperometry. Part 2.2. looks very poor.

ANS: Revised. Added the CV method in Section 2.1.2 is a diagram and details of our experiments using the CV method

6) There is a technical mistake in Table 1: range of lactate mentioned as 9-3.96. Authors should correct it.

 ANS: Revised to 9-39.6.

7)” When the working electrode undergoes an oxidation reaction, the auxiliary electrode undergoes a reduction reaction, and the auxiliary electrode does not participate in the electrochemical reaction but only provides a charge balance function” – meaning is not correct, please change. ANS: Revised on 2.1.3.

8) Area of counter electrode should be at least ten times greater then the area of working electrode. Herein, area of working electrode obviously exceed the area of counter electrode. Authors should explain this point more thoroughly.

ANS: Due to limited size of percutaneous microneedles to penetrate the skin demanded to be minimally invasive, most of the related papers employed three electrodes of similar area,  such as” https://www.sciencedirect.com/science/article/pii/S0956566317302944”,https://onlinelibrary.wiley.com/doi/full/10.1002/adfm.202009850,” https://www.sciencedirect.com/science/article/pii/S0956566317300167 ”,” http://www.electrochemsci.org/papers/vol10/100302455.pdf ”,” https://ieeexplore.ieee.org/abstract/document/8346574”.

9) On Figure 5, counter electrode and reference electrode look like plate with two dark areas, though authors describe it in the manuscript as 3*1 array. Authors should explain this point more thoroughly.

 ANS: WE is 3*3 metal microneedles, CE and RE are 2*1 metal microneedles.

.

10) From Figure 3 it is not clear: if microneedle penetrates through the dermis till the subcutis or it stays in the dermis? It has been written overall the manuscript: “sensor inserted under the skin”, that seems to be a result of surgery. But at Figures 18 and 19 there are two healthy volunteers with microneedles system applied over the body. Authors should operate with terms more accurate.

 ANS: Microneedles penetrate the skin; we make microneedles that are 1 mm long enough to penetrate under the skin to the dermal tissue.

11) The presented idea of lactate biosensor microneedle system is not new. There are many similar publications (https://doi.org/10.1016/j.microc.2020.105830, 10.1016/j.snb.2019.127645, 10.1021/acs.analchem.9b05109), that’s why 17 references is not sufficient.  ANS: Thank you for reminding us to refer to more related papers, which have been added in the reference list. However, we want to stress on the following points:

Our paper provides a low cost, novel microneedle array system and performs human trials. While most of the other papers focus on single-shot measurements, our paper can demonstrate continuous measurements. And the measurement of lactate is rare, basically the other papers are mainly based on drug administration and glucose measurement.

12) Line 199: “Nafion is applied on the enzyme layer to obtain higher electrode surface adsorption and improve electrode detection sensitivity.” Nafion is a polyanion and usually employs in lactate biosensors to exclude the influence of interfering compounds. The disposition of Nafion layer over the recognition layer is also unclear: Nafion layer should repulse the negatively charged lactate anion. Add the explanation of this layer organization.

ANS: Revised on line 190.191.” Nafion has excellent ion exchange properties and biocompatibility and can be used to suppress interference.”

13) The manuscript describes the performance of continuous lactate monitoring system but there is no mention about its analytical or operational characteristics. There is no determination of concentration range, LOD, lifetime, stability and so on. Lack of this information is inadmissible.

14) What does mean “first piece of microneedle” and “second piece of microneedle” on Figure 13? “Two working electrodes were used for these measurements” can be considered that authors made only two attempts to detect lactate. It is not enough for adequate results interpretation. ANS: Revised on Figure 14.

15) How did the composition of antiperspirant affect the results? What about contamination problem?

ANS: “The antiperspirant in this experiment was sprayed on the skin instead of the microneedles and waited for 30 minutes before the experiment. And when spraying antiperspirant, we tried to avoid the place where the microneedle is placed.” Revised on chapter 3.3.

16) If 40 volunteers were tested, why only two results presented in the article? Where the standard deviation for the group?

ANS: The experimental results of 40 volunteers are shown in Figure 22, and there are not only two results.

17) Why do authors not discuss the difference between lactate concentration and its changing rate in blood and subcutaneous tissue fluid? Why is there no information about normal and abnormal levels of lactate in subcutaneous tissue fluid?

ANS: “In this experiment, the lactic acid in the tissue fluid was first measured. In the experiment, it was found that the sensitivity of lactic acid in tissue fluid was higher than that in blood. Blood lactate is mainly used to correct the interstitial fluid lactate measured by our sensor.” Revised on chapter 3.5.

18) There is no information about the pH of the working buffer. It might be critical for enzymatic biosensors. PBS buffer abbreviation means Phosphate Buffered Saline solution – please check if NaCl was present in the media in this work.

ANS: pH=7.2 and there is no NaCl in it .

19) There are some misprints and poor phrases in the manuscript: Line 41 “of relatively high”, Line 29 “CLMS was tested on 40 human subjects was who rode” – second “was” is unnecessary.

ANS: Revised.

20) Figure captions should be extended and contain all the conditions of the experiments. ANS: Revised.

Round 2

Reviewer 1 Report

After revision, the authors did partially improve the manuscript, but there are still some unacceptable mistakes and missing data which still require major revisions. Thus, the specific issues are:

  • Line 45-46: “l” for litre was still not changed to capital “L”, as well as in Fig 22
  • Chapter 2: Material and Methods section – was improved, but still needs further attention.

- List of materials was introduced, but there is still data missing. PBS stands for phosphate buffer solution or phosphate buffer saline solution?

- What is the abbreviation for polyaniline? The accepted convention is PANI.

- PAI is once referred to as “Polyamide-imide(PAI)layer” and once “conductive polymer film (PAI)”. Please define and choose one. The reviewer is under the impression that this PAI is actually the PANI film. Please clarify.

- The authors added the CV profile of the microneedles only in the presence of lactate. The reviewer kindly asked them to also overlay the CV profile in its absence, so that they prove the difference in signal. The authors are also requested to use a graph processing software (excel, origin or any other software) not a print-screen from the Palmsens acquisition software.

  • Line 246: “oxidation of hydrogen oxide” - do the authors mean hydrogen peroxide - H2O2 or -OH, hydroxide? According to the process electrochemistry it should be H2O2, however, this highlights that the authors still don’t completely understand the mechanism presented in Fig 6. I assume the working principle would be based on indirect detection of lactate. This means that with increasing concentration of lactate, H2O2 generation increases, resulting in high current signals (generated though the interaction of LOX with lactate), so the actual signal is given by H2O2. Is this correct? Please clearly explain this under chapter 2.2.2
  • Now line 173-174: the authors did not correct/ comment on this request:” Line 183: what is this oxidized chemical mediator the authors refer to?” Mediators in electrochemical reactions promote electron transfer, which is not the case here. Does mediator stand for something else? The authors should clarify this.
  • Instrumentation for SEM is missing
  • Figure 13: please define the value on the Y axis in the figure itself, not in the Figure caption. Current is denoted with I(uA), not “value (uA)” Also correct on X axis mg/dL- capital “L”. Please move the text “When the concentration is 0, a tiny current signal will be 308 detected because there are still impurities in the PBS.” from the figure caption to the description, after “….different concentrations of lactate.” – line 305
  • Chapter 3.2. -> the authors say it was revised, however there is absolutely no change in the text compared to the first version of the manuscript. The text should be clear for non-specialists also, so please rephrase this chapter. It is unclear how the measurements were performed: how was lactate added during measurement in air? Also, an electrochemical measurement requires an aqueous media, otherwise the electrodes can be compromised, so please explain what exactly happens.
  • Figure14 and 15 could be further improved by replacing “current value” with I
  • Line 329 was not addressed by the authors: “The experimentally measured a was approximately equal to 0.1.” – this sentence is confusing, 0.1 is the value of the correction, the standard deviation? How was it used. If the authors mention it, it should be better explained.
  • Figure 21: correct on Y axis mg/dL- capital “L”

Author Response

    • Line 45-46: “l” for litre was still not changed to capital “L”, as well as in Fig 22

    ANS: Changed to English uppercase

    • Chapter 2: Material and Methods section – was improved, but still needs further attention.

    - List of materials was introduced, but there is still data missing. PBS stands for phosphate buffer solution or phosphate buffer saline solution?

    ANS: Changed to full name. PBS stands for phosphate buffer solution.

    - What is the abbreviation for polyaniline? The accepted convention is PANI.

    ANS: Changed to PANI.

    - PAI is once referred to as “Polyamide-imide(PAI)layer” and once “conductive polymer film (PAI)”. Please define and choose one. The reviewer is under the impression that this PAI is actually the PANI film. Please clarify.

    ANS: Revised to PANI.

    - The authors added the CV profile of the microneedles only in the presence of lactate. The reviewer kindly asked them to also overlay the CV profile in its absence, so that they prove the difference in signal. The authors are also requested to use a graph processing software (excel, origin or any other software) not a print-screen from the Palmsens acquisition software.

    ANS: Revised in Figures 2 by adding the CV profile of the microneedles only and in the presence of lactate in graph processing software of Excel. 

    • Line 246: “oxidation of hydrogen oxide” - do the authors mean hydrogen peroxide - H2O2 or -OH, hydroxide? According to the process electrochemistry it should be H2O2, however, this highlights that the authors still don’t completely understand the mechanism presented in Fig 6. I assume the working principle would be based on indirect detection of lactate. This means that with increasing concentration of lactate, H2O2 generation increases, resulting in high current signals (generated though the interaction of LOX with lactate), so the actual signal is given by H2O2. Is this correct? Please clearly explain this under chapter 2.2.2

    ANS: yes, revised to hydrogen peroxide

    • Now line 173-174: the authors did not correct/ comment on this request:” Line 183: what is this oxidized chemical mediator the authors refer to?” Mediators in electrochemical reactions promote electron transfer, which is not the case here. Does mediator stand for something else? The authors should clarify this.
    • Instrumentation for SEM is missing

    ANS: The brand of SEM is JEOL and the model is JSM-7610F

    • Figure 13: please define the value on the Y axis in the figure itself, not in the Figure caption. Current is denoted with I(uA), not “value (uA)” Also correct on X axis mg/dL- capital “L”. Please move the text “When the concentration is 0, a tiny current signal will be 308 detected because there are still impurities in the PBS.” from the figure caption to the description, after “….different concentrations of lactate.” – line 305

    ANS: The X and Y axes have been changed

    • Chapter 3.2. -> the authors say it was revised, however there is absolutely no change in the text compared to the first version of the manuscript. The text should be clear for non-specialists also, so please rephrase this chapter. It is unclear how the measurements were performed: how was lactate added during measurement in air? Also, an electrochemical measurement requires an aqueous media, otherwise the electrodes can be compromised, so please explain what exactly happens.

    ANS:” As can be seen in Figure 16, because lactic acid was not measured after subcutaneous penetration, the three electrodes were emptied, and no lactic acid was measured, resulting in the continuous superposition of residual current and an increase in current. As can be seen in Figure 117, when lactic acid is measured, the downward signal will be measured because each measurement will lead to the destruction of the enzyme.”

    • Figure14 and 15 could be further improved by replacing “current value” with I

    ANS:Revised

    • Line 329 was not addressed by the authors: “The experimentally measured a was approximately equal to 0.1.” – this sentence is confusing, 0.1 is the value of the correction, the standard deviation? How was it used. If the authors mention it, it should be better explained.

    ANS:Revised on 3.4

    • Figure 21: correct on Y axis mg/dL- capital “L”

    ANS:Revised

Reviewer 2 Report

Although the authors made some changes in the manuscript, main problems remained unsolved. I cannot recommend this manuscript for publishing in the Sensors journal in the present form, because no analytical and operational characteristics of the monitoring system were added.

  • After changes made, normal and abnormal levels of lactate in blood (Line 46, Table 1) are significantly overlapped. What was the area of analytical interest for lactate determination in this manuscript?
  • There is still no explanation of HEMA acronym. PAI was defined as polyamide-imide layer,conductive polymer film (Lines 153, 200). In Part 2.1., authors wrote:”The working electrode is then covered with plating gold to facilitate subsequent material adhesion including polyaniline, lactate enzyme, Nafion and Poly HEMA“. So what the PAI layer means, conductive polyaniline or dielectric polyamide-imide layer?
  • Figure 2: There are no obvious peaks on the cyclic voltammogram attributed to the lactate oxidation. What signal did the authors exactly measure?
  • Authors did not change Lines 149-150:”The reference electrode is a 3*1 array covered with AgCl, and the counter electrode is a 3*1 array covered with plating gold.”
  • Please correct the following phrase:”Lactate oxidase (LOX) is often used as a biological receptor to interact with LOX (???) which specifically recognizes L-lactate and O2 (???) to produce redox chemical reactants which can be measured by electrochemical methods.”
  • Authors did not explain how they overcame the electrostatic repulsion between negatively charged lactate-anion and Nafion layer.
  • I strictly recommend to replace the word combination “insert under the skin” with “penetrate the skin”.
  • I cannot agree with the authors concerning lactate detection rarity: many papers describe it as common as glucose detection. There are many scientific groups in this area of chemistry. Meanwhile I confirm importance of the in vivo testing described in the manuscript.
  • My remark concerning the analytical or operational characteristics of monitoring system remained without response. There is still information on the concentration range, LOD, lifetime, stability, signal drift and so on. Lack of this information is inadmissible.
  • About Figure 14: what impurities are in the working solution? Where is the standard deviation of the signal? What is the lactate detection range? What is the sensitivity of the detection? What is the lactate limit of detection? Please adduce this information.
  • If antiperspirant was not sprayed on the microneedles place, how could it affect the signal (Figure 18)?
  • According to the authors’ answer: ”sensitivity of lactic acid in tissue fluid was higher than that in blood”. Sensitivity is the slope of the calibration plot. According to the Figure 21, it depends on experimental period.
  • If there were no NaCl in the buffer solution, authors should not use the PBS acronym (to be replaced with PB – phosphate buffer).
  • Figure caption should be more formal.

Author Response

  • After changes made, normal and abnormal levels of lactate in blood (Line 46, Table 1) are significantly overlapped. What was the area of analytical interest for lactate determination in this manuscript?

ANS: We want to measure the value from the value without movement to the value after movement (>10) is what we want

  • There is still no explanation of HEMA acronym. PAI was defined as polyamide-imide layer,conductive polymer film (Lines 153, 200). In Part 2.1., authors wrote:”The working electrode is then covered with plating gold to facilitate subsequent material adhesion including polyaniline, lactate enzyme, Nafion and Poly HEMA“. So what the PAI layer means, conductive polyaniline or dielectric polyamide-imide layer?

ANS: HEMA is 2-hydroxyethyl methacrylate.PAI has been changed to PANI(polyamide-imide layer)

  • Figure 2: There are no obvious peaks on the cyclic voltammogram attributed to the lactate oxidation. What signal did the authors exactly measure?

ANS:Changed to figure2.and figure3.

  • Authors did not change Lines 149-150:”The reference electrode is a 3*1 array covered with AgCl, and the counter electrode is a 3*1 array covered with plating gold.”

ANS:Revised to “The reference electrode is a 2*1 array covered with AgCl, and the counter electrode is a 2*1 array covered with plating gold.”

  • Please correct the following phrase:”Lactate oxidase (LOX) is often used as a biological receptor to interact with LOX (???) which specifically recognizes L-lactate and O2 (???) to produce redox chemical reactants which can be measured by electrochemical methods.”

ANS:Revised

  • Authors did not explain how they overcame the electrostatic repulsion between negatively charged lactate-anion and Nafion layer.

ANS: Nafion can mainly repel impurities, and will not cause the problem that lactic acid cannot be measured, and Nafion has a large hole, so the problem of repulsion is small.

  • I strictly recommend to replace the word combination “insert under the skin” with “penetrate the skin”.

ANS:Revised

  • I cannot agree with the authors concerning lactate detection rarity: many papers describe it as common as glucose detection. There are many scientific groups in this area of chemistry. Meanwhile I confirm importance of the in vivo testing described in the manuscript.

ANS: Thank you for your attention to human trials. Our experiments not only include human trials, but also complete low-cost manufacturing of our sensors. It's different from others with new ideas.

  • My remark concerning the analytical or operational characteristics of monitoring system remained without response. There is still information on the concentration range, LOD, lifetime, stability, signal drift and so on. Lack of this information is inadmissible.

ANS: After many experiments and the process of making microneedles, we learned that the sensor we made can measure about 5mg/dL to 360mg/dl. And each microneedle manufactured will have a measurement difference of about 4-7uA. The service life can be about one month.

  • About Figure 14: what impurities are in the working solution? Where is the standard deviation of the signal? What is the lactate detection range? What is the sensitivity of the detection? What is the lactate limit of detection? Please adduce this information.

ANS: Revised on Figure 15.

  • If antiperspirant was not sprayed on the microneedles place, how could it affect the signal (Figure 18)?

ANS: The antiperspirant is not sprayed on the microneedle, but is sprayed near the skin, as far as possible not to affect the insertion of the microneedle.

  • According to the authors’ answer”sensitivity of lactic acid in tissue fluid was higher than that in blood”. Sensitivity is the slope of the calibration plot. According to the Figure 21, it depends on experimental period.

ANS: Because earlobe blood is taken by the blood measurement method, the lactic acid concentration measured in the local interstitial fluid must be faster than the earlobe blood.

  • If there were no NaCl in the buffer solution, authors should not use the PBS acronym (to be replaced with PB – phosphate buffer).

ANS:Revised

Round 3

Reviewer 1 Report

The authors significantly improved the manuscript this time, however, there are still some issues and enquiries the authors did not answer from previous revisions:

  • Line 43: continuous measurement, not continuously
  • Lines 132-133: the authors must add at which potential they read the value of the oxidation current. This potential must be clearly established and kept throughout all measurements. Is it the potential of 0.35 V specified in line 271? The reviewer hopes the answer is yes, otherwise consistency issues on the measurement methodology arise.
  • Line 139: correct PalmSens machine to potentiostat
  • Line 162: Why does PANIs porosity matter? The authors put emphasis on it, so it should be explained.
  • Line 181: the authors did still not answer initial enquiry: - “Line 183: what is this oxidized chemical mediator the authors refer to and how did they react LOX with it for regeneration? This needs an explanation!
  • Lines 187 and 189: it is hydrogen peroxide – H2O2, not hydrogen oxide! Hydrogen oxide is the simple systematic name for water, H2O.
  • Chapters 2.1.6 and 2.1.7 can be put under chapter 2.1.6, keeping its title
  • Figure 7 – correct PAI to PANI in the diagram
  • Line 224: change auxiliary to CE – keep consistency throughout the manuscript
  • Line 258: change auxiliary to counter – keep consistency
  • Line 306: add “solutions with” to “5 different solutions with lactate concentrations …”
  • Line 308: replace zones with electrolytes
  • Line 314: can “many experiments” be quantified? e.g. 10 sets of measurements. It would also benefit the manuscript to add an error interval to the 5-360 mg/dL range.
  • Line 316: change uA to μA
  • Line 317: PBS not PB
  • Line 325-326: delete whole sentence from figure caption: “When the concentration ….” It was already mentioned in the text above
  • Line 339: I think the authors meant figure 16
  • Line 399: correct “A total of ….”

In conclusion, it is the reviewer's opinion that the manuscript still needs some revisions to be accepted for publication in Sensors. However, the reviewer appreciates the effort the authors made in order to improve this manuscript!

Author Response

  • Line 43: continuous measurement, not continuously

ANS:Revised.

  • Lines 132-133: the authors must add at which potential they read the value of the oxidation current. This potential must be clearly established and kept throughout all measurements. Is it the potential of 0.35 V specified in line 271? The reviewer hopes the answer is yes, otherwise consistency issues on the measurement methodology arise.

ANS:Revised. The experiment is to read the position of 0.35V

  • Line 139: correct PalmSens machine to potentiostat

ANS:Revised.

  • Line 162: Why does PANIs porosity matter? The authors put emphasis on it, so it should be explained.

ANS:Revised.” The porosity of PANI is important because the more pores there are, the more enzymes can be adsorbed on it.”

  • Line 181: the authors did still not answer initial enquiry: - “Line 183: what is this oxidized chemical mediator the authors refer to and how did they react LOX with it for regeneration? This needs an explanation!

ANS:Revised.

  • Lines 187 and 189:it is hydrogen peroxide – H2O2, not hydrogen oxide! Hydrogen oxide is the simple systematic name for water, H2O.

ANS:Revised.

  • Chapters 2.1.6 and 2.1.7 can be put under chapter 2.1.6, keeping its title

ANS:Revised.

  • Figure 7 – correct PAI to PANI in the diagram

ANS:Revised.

  • Line 224: change auxiliary to CE – keep consistency throughout the manuscript

ANS:Revised.

  • Line 258: change auxiliary to counter – keep consistency

ANS:Revised.

  • Line 306: add “solutions with” to “5 different solutions with lactate concentrations …”

ANS:Revised.

  • Line 308: replace zones with electrolytes

ANS:Revised.

  • Line 314: can “many experiments” be quantified? e.g. 10 sets of measurements. It would also benefit the manuscript to add an error interval to the 5-360 mg/dL range.

ANS:Revised. We did about 20 experiments.

  • Line 316: change uA to μA

ANS:Revised.

  • Line 317: PBS not PB

ANS:Revised.

  • Line 325-326: delete whole sentence from figure caption: “When the concentration ….” It was already mentioned in the text above

ANS:Revised.

  • Line 339: I think the authors meant figure 16

ANS:Revised.

  • Line 399: correct “total of ….”

ANS:Revised.

Reviewer 2 Report

Authors significantly improved the manuscript, so I would recommend accepting the manuscript after minor changes:

1) Line 75: “LDH biomarker” should be changed to “LDH enzyme”.

2) Line 70: duplication “can be can be”.

3) Line 176: Phrase “L-lactic can be electrochemically measured in terms of the level of L-lactic oxidase” should be changed to “L-lactate can be electrochemically measured in terms of the LOX activity“.

4) Line 177: “The electrochemical working principle is a biosensor that uses an electrochemical reaction to detect the lactate concentration” should be rephrased as “We have applied biosensor based on electrochemical reaction to determinate the lactate concentration”.

5) Lines 187, 189: “hydrogen oxide” was used instead of “hydrogen peroxide”.

6) On Figure 7, “PAI” was used instead of “PANI”.

7) Section 2.1.5 should be thoroughly checked. Authors were discussed the mediator behavior, but what was used as mediator? There were no mediators mentioned in Figure 6. Please adjust in accordance with your experiment conditions.

Also I would recommend to change the Figure 6 caption to “Signal generation scheme”.

8) Lines 183, 255: Electric ions do not exist, electrons can not adsorb at conductive surface.

9) Line 296: I would recommend to substitute the phrase “Sensor effectiveness experiment” for ‘Sensor performance experiment”. Signal increase should be clarified.

10) Line 317: The impurities should not be explained with buffer components.

11) Line 326: “PBS” was used instead of “PB”.

12) Line 316: Explain what measurement variation is, and how many microneedles were taken for this assessment? 10 uI corresponds to 10% deviation of signal. Would this large deviation be an obstacle for sensor performance?

13) Line 339: “Figure 117” should be checked.

14) Line 330: Phrase “lactate concentration increased gradually” should be rephrased as “signal increased gradually”.

15) Section 3.2. should be rewritten to clarified the experiment and whether signal or lactate concentration was actually changed.

16) Authors mentioned that “sensor was invalidated in air”. Did they work without any buffer solution containing lactate?

17) Line 340: What did authors mean in phrase” deteriorates the enzyme”?

18) Line 358: Phrase “instead of the microneedles” should be rephrased.

19) Lines 399-402: duplication with the Table 2 should be removed.

20) Lines 411, 412: Phrase “that the lactate enzyme on the microneedle sensor” should be changed to “that the lactate signal on the microneedle sensor”.

Author Response

Line 75: “LDH biomarker” should be changed to “LDH enzyme”.

ANS:Revised.

Line 70: duplication “can be can be”.

ANS:Revised.

Line 176: Phrase “L-lactic can be electrochemically measured in terms of the level of L-lactic oxidase” should be changed to “L-lactate can be electrochemically measured in terms of the LOX activity“.

ANS:Revised.

Line 177: “The electrochemical working principle is a biosensor that uses an electrochemical reaction to detect the lactate concentration” should be rephrased as “We have applied biosensor based on electrochemical reaction to determinate the lactate concentration”.

ANS:Revised.

Lines 187, 189: “hydrogen oxide” was used instead of “hydrogen peroxide”.

ANS:Revised.

On Figure 7, “PAI” was used instead of “PANI”.

ANS:Revised.

7) Section 2.1.5 should be thoroughly checked. Authors were discussed the mediator behavior, but what was used as mediator? There were no mediators mentioned in Figure 6. Please adjust in accordance with your experiment conditions.

Also I would recommend to change the Figure 6 caption to “Signal generation scheme”.

ANS:Revised.

Lines 183, 255: Electric ions do not exist, electrons can not adsorb at conductive surface.

ANS:Revised.

Line 296: I would recommend to substitute the phrase “Sensor effectiveness experiment” for ‘Sensor performance experiment”. Signal increase should be clarified.

ANS:Revised.

Line 317: The impurities should not be explained with buffer components.

ANS:Revised.” When the concentration is 0, because PBS has impedance, when we apply voltage to PBS, we still get a small current value.”

Line 326: “PBS” was used instead of “PB”.

ANS:Revised.

Line 316: Explain what measurement variation is, and how many microneedles were taken for this assessment? 10 uI corresponds to 10% deviation of signal. Would this large deviation be an obstacle for sensor performance?

ANS:Revised. The resulting discrepancy can be modified with firmware at a later stage and will not cause difficulties with the sensor.

Line 339: “Figure 117” should be checked.

ANS:Revised.

Line 330: Phrase “lactate concentration increased gradually” should be rephrased as “signal increased gradually”.

ANS:Revised.

Section 3.2. should be rewritten to clarified the experiment and whether signal or lactate concentration was actually changed.

ANS:Revised.

 Authors mentioned that “sensor was invalidated in air”. Did they work without any buffer solution containing lactate?

ANS:Revised. Place the microneedle in the AGAR, not air.

Line 340: What did authors mean in phrase” deteriorates the enzyme”?

ANS:Revised.

Line 358: Phrase “instead of the microneedles” should be rephrased.

ANS:Revised.

Lines 399-402: duplication with the Table 2 should be removed.

ANS:Revised.

Lines 411, 412: Phrase “that the lactate enzyme on the microneedle sensor” should be changed to “that the lactate signal on the microneedle sensor”.

ANS:Revised.
